# Association of Kawasaki disease with urbanization level and family characteristics in Taiwan: A nested case–control study using national-level data

Chung-Fang Tseng[1], Hsiao-Chen Lin[1], Chung-Yuh Tzeng[2], Jing-Yang Huang[3], Chih-Jung Yeh[1]*, James Cheng-Chung Wei[3,4,5,6]*

1 School of Public Health, Chung Shan Medical University, Taichung, Taiwan, 2 Department of Orthopedics, Taichung Veterans General Hospital, Taichung, Taiwan, 3 Institute of Medicine, Chung Shan Medical University, Taichung, Taiwan, 4 Division of Allergy, Immunology and Rheumatology, Chung Shan Medical University Hospital, Taichung, Taiwan, 5 Graduate Institute of Integrated Medicine, China Medical University, Taichung, Taiwan, 6 Department of Medical Research, Taichung Veterans General Hospital, Taichung, Taiwan

* alexyeh@csmu.edu.tw (CJY); jccwei@gmail.com (JCCW)

**Data Availability Statement:** Data are not publicly available in order to protect the privacy of research participants. Datasets from the National Health

## Abstract

Kawasaki disease (KD) is an inflammatory vasculitis disorder of unknown etiology. It is a rare but fatal disease and the leading cause of acquired coronary heart disease in children under the age of 5 years. We examined the association of KD with the demographics of family members, parents' characteristics, and perinatal factors in Taiwanese children. This nested case–control study used data from Taiwan's Health and Welfare Data Science Center and initially included children born in Taiwan between January 1, 2006, and December 31, 2015 (n = 1,939,449); the children were observed for KD development before the age of 5 years (n = 7870). The control group consisted of children without KD who were matched with each KD case by sex and birth date at a ratio of 8:1. The odds ratio (ORs) of the aforementioned associations were estimated using conditional logistic regression. The risk of KD decreased in children with younger parents [<25 years; younger maternal age, OR = 0.72, 95% confidence interval (CI), 0.66–0.79; younger paternal age, OR = 0.68, 95% CI, 0.59–0.78], lower socioeconomic status, more than 2 siblings (OR = 0.80, 95% CI, 0.73–0.89), and siblings with a history of KD (OR = 4.39, 95% CI, 3.29–5.86). Children living in suburban (OR = 0.95, 95% CI, 0.90–1.00) and rural (OR = 0.81, 95% CI, 0.74–0.90) areas exhibited a lower risk of KD than children living in urban areas. In conclusion, a higher incidence rate of KD was observed in children aged <5 years who had an urban lifestyle, had siblings with KD, were born to older mothers, and belonged to high-income and smaller families. Parental allergic or autoimmune diseases were not associated with the risk of KD.

## Introduction

Kawasaki disease (KD) is an inflammatory vasculitis disorder of unknown etiology [1]. The major clinical manifestations of KD include mucosal changes, conjunctivitis, polymorphous

Insurance Research Datasets are available through a request to the Health and Welfare Data Science Center (HWDSC). The application number of this study is H108158, registered with HWDSC. Contact details for data access requests: Name: Da-Lun Wu (吳岱倫) Email: STDLWU@mohw.gov.tw.

**Funding:** The Institutional Review Board of Chung Shan Medical University Hospital approved the study (CS19009). The funders had no role in study design, data collection and analysis, decision to publish, or preparation of the manuscript.

**Competing interests:** The authors have declared that no competing interests exist.

rash, extremity changes, and lymphadenopathy [1]. KD is a rare but fatal disease and the leading cause of acquired coronary heart disease in children under the age of 5 years [2]. KD was first reported in the 1960s [3], and the majority of cases have been reported in East Asia [4]. Genetics, immune-mediated inflammatory process, and environmental or infectious triggers contribute to KD development [1]. Parental characteristics, especially maternal characteristics, can predict the risk of KD [5–7]. Maternal autoimmune disease might induce an epigenetic predisposition to KD in offspring [5]. Maternal age is significantly associated with coronary artery lesion (CAL) formation in offspring with KD, and maternal age of less than 32 years reduces the incidence of CAL formation in offspring with KD [6]. A family history of allergy is significantly higher in children with KD [8]. Higher household income, smaller family size, and urbanization at birth are associated with increased KD incidence [9–12]. Therefore, we confirmed the etiological association of KD with the demographics of family members, parents' characteristics, and perinatal and environmental factors in the real world. This case–control study investigated the parental factors and diseases responsible for KD development by using data from the National Health Insurance Research Database (NHIRD) and National Birth Reporting Database (NBRD).

## Materials and methods

### Data source

For this retrospective nested case–control study, we selected children born in Taiwan between January 1, 2006, and December 31, 2015, (n = 1,939,449) and observed the development of KD before the age of 5 years in these children. We used data from Taiwan's Health and Welfare Data Science Center, which maintains many nationwide databases in Taiwan for assisting academic research [13]. We used three nationwide databases: the NHIRD (from 2003 to 2017), NBRD (from 2006 to 2015), and National Death Index Database (NDID, from 2006 to 2017). From the databases, baseline characteristics, namely demographic characteristics, parental allergic diseases, and perinatal conditions, and KD development in offspring, were obtained. The NHIRD is one of the largest and most comprehensive healthcare databases in the world [13] and includes medical claims data of >99% of Taiwan residents. Data on medical claims for inpatient and outpatient visits to clinics or hospitals (diagnosis codes and date of visit) were obtained from the NHIRD [13]. The incidence of maternal diseases (hypertension and diabetes) and childhood KD was obtained by reviewing medical claim records. We also extracted data on the child's date of birth and parental monthly income. Data on the characteristics of mothers, infants, and mothers' deliveries were obtained from the NBRD. We also extracted data on the delivery mode, infant's sex, birth weight, and gestational age. Data on the survival status of infants were obtained from the NDID. Advanced maternal age (≥35 years) has been associated with higher risks of maternal morbidity, obstetric interventions, and adverse pregnancy outcomes [14]. A study demonstrated that children born to mothers aged <25 or >35 years had poorer outcomes in terms of mortality, self-reported health, height, obesity, and the number of diagnosed conditions than those born to mothers aged 25–34 years [15]. Therefore, we categorized maternal/paternal age into four groups: <25, 25–34, 35–45, and ≥45 years old.

All residents in Taiwan have a unique personal identification number, which permits us to link information across these nationwide databases. The relationship between parents and children was confirmed through data linkage with the birth registration system in Taiwan, and the details can be obtained from previous reports [16]. Siblings were identified by matching the child's ID with the mother's ID.

The Institutional Review Board of Chung Shan Medical University Hospital approved the study (CS19009). All data in this study were encrypted and were anonymous during data analysis.

## Definition of children with KD and matched controls

Childhood KD occurring before the age of 5 years was ascertained on the basis of clinicians' use of *International Classification of Diseases, Ninth [Tenth] Revision, Clinical Modification* (*ICD-9-CM* code 446.1 or *ICD-10-CM* code M30.3) diagnosis codes [17]. KD cases were defined when KD diagnosis was recorded in at least one hospital admission or in at least 3 ambulatory clinic visits. According to pediatricians' records, 93.96% of visits were of children with KD. We defined the index date as the date of the first record of KD case.

Initially, we identified 1,939,449 children from our datasets. Of them, we excluded 42,396 children with missing data in NBRD, 610 stillbirth cases, 57,976 multiple birth cases, and 114, 594 children with foreign nationality of the mother. Finally, we included 1,723,873 children; of them, 7870 children developed KD before the age of 5 years, and 1,716,003 children were without KD before the age of 5 years. We adopted the nested case–control study design and selected 8 controls that were matched with each KD case by sex and birth date (within 6 months). In the final analysis, we included 7870 children with KD and 62,960 children without KD (controls; Fig 1).

## Definition of demographics, parental allergic disease, and other covariates at baseline

The baseline period for assessing parental disease was 2 years before the index date. The incidence of parental atopic dermatitis (*ICD-9-CM* code 691), allergic rhinitis (*ICD-9-CM* code 477), and asthma (*ICD-9-CM* code 493) was identified using the outpatient and inpatient records available in the NHIRD. To improve the positive predictive probability of allergic disease, the diagnosis was defined as at least one hospital admission or at least 3 ambulatory clinic visits.

We included covariates to account for the potential confounding effect. The covariates included delivery mode; baseline parental age; insurance type; urbanization; comorbidities [including hypertension (*ICD-9-CM* codes 401–405); preeclampsia (*ICD-9-CM* code 642); hyperlipidemia (*ICD-9-CM* codes 270.0–272.4); chronic liver disease (*ICD-9-CM* code 571); chronic kidney disease (*ICD-9-CM* code 585); gestational diabetes mellitus or chronic diabetes (*ICD-9-CM* code 250); chronic obstructive pulmonary disease (COPD; *ICD-9-CM* codes 491, 492, and 496); autoimmune disease (*ICD-9-CM* codes 710, 714, and 720); and depressive disorder (*ICD-9-CM* codes 292 and 293)]; the child's birth year, sex, and birth weight; gestational age; 1-min Apgar score; the number of siblings; and siblings with KD (Table 1).

## Data analysis

We constructed the conditional logistic regression model to explore the association between baseline parental allergic disease and the risk of KD in children before the age of 5 years [18]. We further constructed adjusted models additionally including the child's birth year, fetal sex, paternal age, maternal age, urbanization level, insurance type, maternal comorbidity, birth weight, gestational age, 1-min Apgar score, and the number of siblings to estimate the independent effect of each exposure variable. We included the interaction term to estimate the potential interaction effect between maternal and paternal disease on the risk of KD. All statistical analyses were performed using SAS 9.4 (SAS Institute, Cary, NC, USA). A two-sided *P* value of < .05 indicated statistical significance.

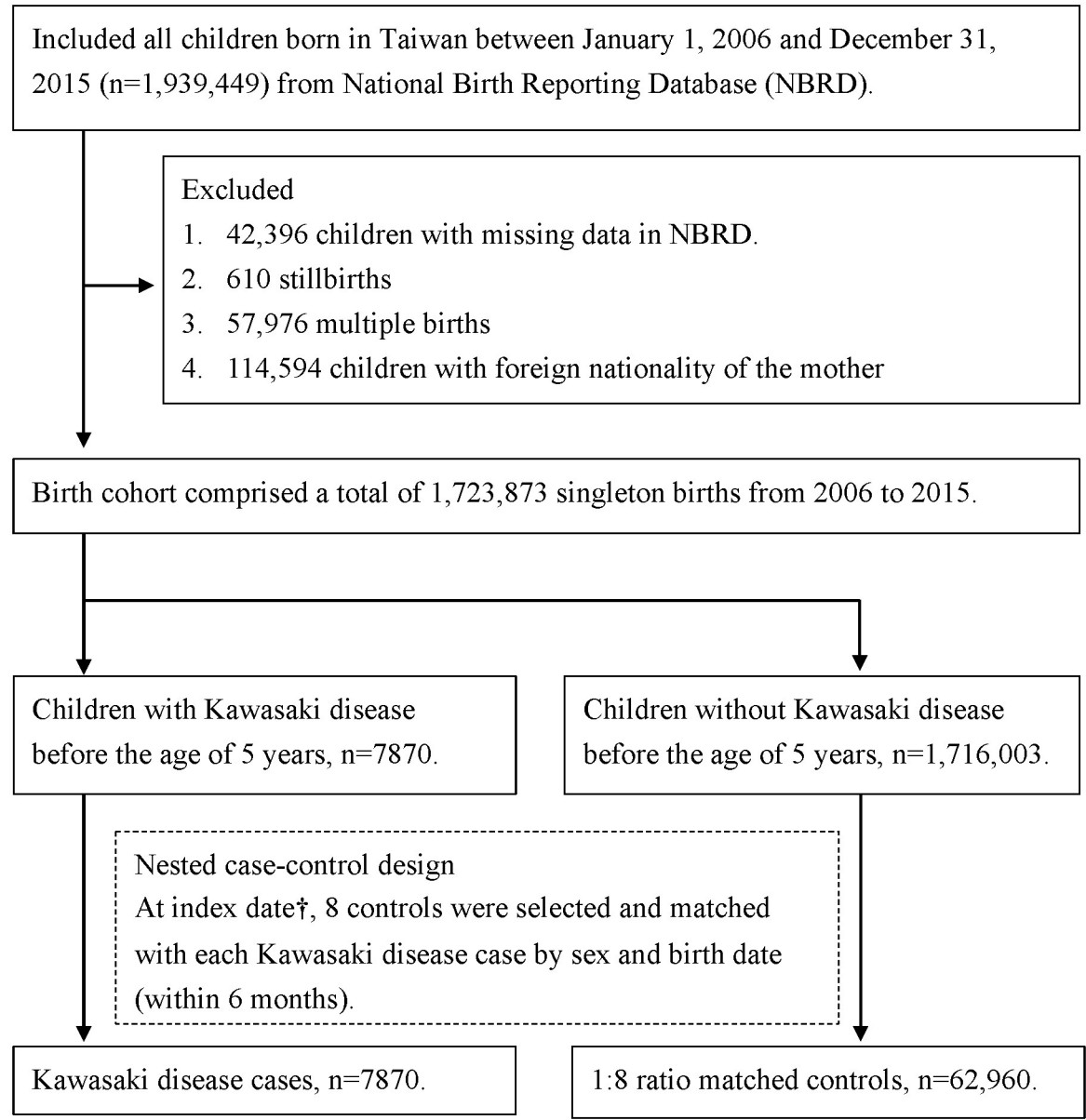

**Fig 1. Flowchart of sample selection.** † Index date was defined as the date when the first Kawasaki disease case was recorded. For control, the same index date was assigned as that of the paired Kawasaki case.

## Results

A total of 7870 children with KD were included, and the incidence rate of KD in children aged below 5 years between 2006 and 2015 ranged from 0.43% to 0.50% per year (Table 1). The incidence rate of KD was the highest in 2013 (0.50%) and the lowest in 2015 (0.43%). The incidence rate was not correlated with the number of births per year; however, in 2015, the highest number of births (191,856) with the lowest incidence rate (0.43%) was recorded. In most children, KD was diagnosed between the ages of 1 and 2 years, followed by 2 and 3 years, 0 and 1 years, 3 and 4 years, and 4 and 5 years, regardless of sex (Tables 1 and 2). Among the included children, 4741 (60.24%) were boys and 3129 (39.76%) were girls (Table 1).

**Table 1. Baseline characteristics of children with Kawasaki disease and without Kawasaki disease before the age of 5 years.**

| Factor | Control Group (N = 62 960) | Kawasaki (N = 7870) | Crude OR | p |
|---|---|---|---|---|
| **Year** | | | | |
| 2006 | 6057(9.62%) | 745(9.47%) | | |
| 2007 | 6212(9.87%) | 790(10.04%) | | |
| 2008 | 6058(9.62%) | 758(9.63%) | | |
| 2009 | 6201(9.85%) | 773(9.82%) | | |
| 2010 | 4805(7.63%) | 619(7.87%) | | |
| 2011 | 6215(9.87%) | 757(9.62%) | | |
| 2012 | 7294(11.59%) | 895(11.37%) | | |
| 2013 | 6638(10.54%) | 866(11.00%) | | |
| 2014 | 6750(10.72%) | 850(10.80%) | | |
| 2015 | 6730(10.69%) | 817(10.38%) | | |
| **Infant's gender** | | | | |
| Male | 37928(60.24%) | 4741(60.24%) | | |
| Female | 25032(39.76%) | 3129(39.76%) | | |
| **Birthweight** | | | | |
| <2500 | 3907(6.21%) | 499(6.34%) | 1.022(0.928–1.126) | 0.6606 |
| 2500–3499 | 49064(77.93%) | 6133(77.93%) | Reference | |
| > = 3500 | 9989(15.87%) | 1238(15.73%) | 0.992(0.929–1.058) | 0.7965 |
| **Gestational weeks** | | | | |
| <36 weeks | 2012(3.20%) | 267(3.39%) | 1.061(0.932–1.208) | 0.3718 |
| 36–41 weeks | 59196(94.02%) | 7404(94.08%) | Reference | |
| Posterm (> = 41 weeks) | 1752(2.78%) | 199(2.53%) | 0.908(0.783–1.054) | 0.2037 |
| **Mode of delivery** | | | | |
| Vaginal delivery | 41015(65.14%) | 5051(64.18%) | Reference | |
| Cesarean delivery | 21945(34.86%) | 2819(35.82%) | 1.043(0.993–1.096) | 0.0893 |
| **Maternal age at delivery** | | | | |
| <25 | 6254(9.93%) | 570(7.24%) | 0.720(0.658–0.788) | < .0001 |
| 25–34 | 43412(68.95%) | 5500(69.89%) | Reference | |
| 35–45 | 13237(21.02%) | 1791(22.76%) | 1.068(1.009–1.130) | 0.0232 |
| > = 45 | 57(0.09%) | 9(0.11%) | 1.246(0.617–2.519) | 0.5393 |
| **Father's age at delivery (miss = 3455)** | | | | |
| <25 | 2408(4.03%) | 203(2.68%) | 0.677(0.585–0.784) | < .0001 |
| 25–34 | 35065(58.63%) | 4364(57.68%) | Reference | |
| 35–45 | 21038(35.18%) | 2850(37.67%) | 1.088(1.035–1.145) | 0.0009 |
| > = 45 | 1298(2.17%) | 149(1.97%) | 0.922(0.776–1.096) | 0.3583 |
| **Insurance** | | | | |
| Public insurance | 4348(6.91%) | 604(7.67%) | 1.070(0.979–1.170) | 0.1343 |
| Labor insurance | 46870(74.44%) | 6082(77.28%) | Reference | |
| Farmer's, fishmen's associated insured | 3260(5.18%) | 355(4.51%) | 0.839(0.750–0.939) | 0.0023 |
| Middle-income to low income family insurance | 168(0.27%) | 11(0.14%) | 0.505(0.274–0.929) | 0.0281 |
| Township office insurance | 6539(10.39%) | 577(7.33%) | 0.680(0.622–0.743) | < .0001 |
| Other | 1775(2.82%) | 241(3.06%) | 1.046(0.912–1.200) | 0.5176 |
| **Urbanization** | | | | |
| Urban | 39180(62.23%) | 5045(64.10%) | Reference | |
| Sub-urban | 18898(30.02%) | 2314(29.40%) | 0.951(0.903–1.002) | 0.0588 |
| Rural | 4882(7.75%) | 511(6.49%) | 0.813(0.739–0.895) | < .0001 |
| **Preeclampsia** | 701(1.11%) | 80(1.02%) | 0.912(0.723–1.151) | 0.4379 |

*(Continued)*

**Table 1.** (Continued)

| Factor | Control Group (N = 62 960) | Kawasaki (N = 7870) | Crude OR | p |
|---|---|---|---|---|
| Maternal preeclampsia | 1050(1.67%) | 135(1.72%) | 1.029(0.859–1.233) | 0.7534 |
| 1-minute apar score <8 | 4569(7.26%) | 593(7.53%) | 1.042(0.953–1.139) | 0.3646 |
| Number of siblings | | | | |
| 0 | 28179(44.76%) | 3591(45.63%) | Reference | |
| 1 | 30008(47.66%) | 3792(48.18%) | 0.992(0.945–1.041) | 0.7333 |
| ≧2 | 4773(7.58%) | 487(6.19%) | 0.801(0.725–0.885) | < .0001 |
| Siblings with KD[a] | 135(0.34%) | 72(1.86%) | 4.392(3.294–5.856) | < .0001 |

[a]only analyzed in children with sibling data.

Table 1 provides the baseline characteristics of children with KD and children without KD. No statistically significant difference in gestational age, delivery mode, maternal preeclampsia, the incidence of gestational diabetes mellitus, and 1-min Apgar score was observed between the KD and control groups. The incidence of KD decreased in children with younger parents [<25 years; maternal odds ratio (OR) = 0.720; 95% confidence interval (CI), 0.658–0.788; $P < .001$ and paternal OR = 0.677; 95% CI, 0.585–0.784; $P < .001$], lower socioeconomic status (township office insured OR = 0.680; 95% CI, 0.622–0.743; $P < .001$), living in rural areas (OR = 0.813; 95% CI, 0.739–0.895; $P < .0001$), and having more than 2 siblings (OR = 0.801; 95% CI, 0.725–0.885; $P < .001$). The incidence of KD increased when siblings had a history of KD (OR = 4.392; 95% CI, 3.294–5.856; $P < .001$). Health insured units reflected parents' socioeconomic status. The decline in the incidence of KD from children with farmers and fishermen insurance (OR = 0.839; 95% CI, 0.750–0.939; $P = .0023$) to children with low income insurance (OR = 0.505; 95% CI, 0.274–0.929; $P = .0281$) to children with township office insurance (OR = 0.680; 95% CI, 0.622–0.743; $P < .001$) was compatible with the decline in the economic status of middle-income family to low-income family. The incidence rate of KD differed by the area of residence, with suburban and rural areas associated with lower incidence of KD (OR = 0.951; 95% CI, 0.903–1.002; $P = .0588$ and OR = 0.813; 95% CI, 0.739–0.895; $P < .001$, respectively) than urban areas. Moreover, families with 2 or more children exhibited a lower incidence of KD (OR = 0.801; 95% CI, 0.725–0.885; $P < .001$) than did families with a single child.

We analyzed parents' diseases as risk factors for KD (Table 2). We excluded children with missing paternal data; thus, a total number of 7566 children with KD were included in the analysis. The parents' underlying conditions included allergic diseases (atopic dermatitis, allergic rhinitis, and asthma), hypertension, hyperlipidemia, diabetes, COPD, depressive disorder, and autoimmune diseases (diffuse connective tissue diseases, rheumatoid arthritis and other inflammatory polyarthropathies, and ankylosing spondylitis and other inflammatory spondylopathies). The crude ORs indicated significant associations of paternal allergic rhinitis, hypertension, hyperlipidemia, and chronic liver disease with offspring KD, and the values of the adjusted ORs were similar to the crude ORs. Parents' allergic diseases were not related to the number of offspring with KD. The adjusted ORs of hypertension, hyperlipidemia, and chronic liver disease (paternal chronic diseases) were 1.137 (95% CI, 0.987–1.309; $P = .0748$), 1.125 (95% CI, 0.991–1.277; $P = .0700$), and 1.085 (95% CI, 0.988–1.192; $P = .0861$), respectively. No significant differences in parents' autoimmune diseases were observed between the KD and control groups.

**Table 2. Results of conditional multivariable logistic regression analysis for risk factors among children with Kawasaki disease.**

| Factor | Control Group (N = 59809) | Kawasaki (N = 7566) | Crude OR (95% CI) | Adjusted OR (95% CI) |
|---|---|---|---|---|
| **Allergy disease** | | | | |
| **Atopic dermatitis** | | | | p for interaction = 0.2668 |
| Mother | 1119(1.87%) | 141(1.86%) | 0.996(0.835–1.189) | 0.991(0.828–1.186) |
| Father | 303(0.51%) | 37(0.49%) | 0.965(0.686–1.359) | 0.905(0.630–1.300) |
| **Allergic rhinitis** | | | | p for interaction = 0.7031 |
| Mother | 5044(8.43%) | 647(8.55%) | 1.015(0.932–1.106) | 1.003(0.913–1.102) |
| Father | 4536(7.58%) | 622(8.22%) | 1.092(1.000–1.191) | 1.072(0.973–1.181) |
| **Asthma** | | | | p for interaction = 0.3905 |
| Mother | 871(1.46%) | 123(1.63%) | 1.118(0.924–1.353) | 1.089(0.893–1.329) |
| Father | 834(1.39%) | 100(1.32%) | 0.947(0.769–1.167) | 0.896(0.719–1.117) |
| **Hypertension** | | | | p for interaction = 0.2078 |
| Mother | 606(1.01%) | 83(1.10%) | 1.048(0.891–1.234) | 1.127(0.882–1.441) |
| Father | 1642(2.75%) | 238(3.15%) | 1.151(1.002–1.321) | 1.137(0.987–1.309) |
| **Hyperlipidemia** | | | | p for interaction = 0.5155 |
| Mother | 385(0.64%) | 42(0.56%) | 0.862(0.626–1.186) | 0.873(0.620–1.229) |
| Father | 2025(3.39%) | 294(3.89%) | 1.154(1.018–1.307) | 1.125(0.991–1.277) |
| **Chronic liver disease** | | | | p for interaction = 0.4180 |
| Mother | 1268(2.12%) | 168(2.22%) | 1.048(0.891–1.234) | 1.044(0.872–1.251) |
| Father | 4146(6.93%) | 575(7.60%) | 1.104(1.009–1.209) | 1.085(0.988–1.192) |
| **Diabetes** | | | | p for interaction = 0.6786 |
| Mother | 690(1.15%) | 76(1.00%) | 0.869(0.685–1.103) | 0.850(0.666–1.085) |
| Father | 693(1.16%) | 90(1.19%) | 1.027(0.823–1.281) | 1.028(0.821–1.288) |
| **COPD** | | | | p for interaction = 0.3199 |
| Mother | 241(0.40%) | 35(0.46%) | 1.149(0.805–1.639) | 1.065(0.730–1.554) |
| Father | 387(0.65%) | 52(0.69%) | 1.063(0.795–1.421) | 1.001(0.740–1.355) |
| **Autoimmune disease** | | | | |
| **Diffuse diseases of connective tissue** | | | | p for interaction = 0.9299 |
| Mother | 341(0.57%) | 43(0.57%) | 0.997(0.725–1.370) | 0.971(0.705–1.336) |
| Father | 86(0.14%) | 10(0.13%) | 0.919(0.477–1.770) | 0.957(0.495–1.853) |
| **Rheumatoid arthritis and other inflammatory polyarthropathies** | | | | p for interaction = 0.9403 |
| Mother | 70(0.12%) | 11(0.15%) | 1.243(0.658–2.347) | 1.230(0.649–2.331) |
| Father | 67(0.11%) | 13(0.17%) | 1.535(0.847–2.781) | 1.530(0.842–2.782) |
| **Ankylosing spondylitis and other inflammatory spondylopathies** | | | | p for interaction = 0.9192 |
| Mother | 101(0.17%) | 14(0.19%) | 1.096(0.626–1.918) | 1.105(0.630–1.938) |
| Father | 329(0.55%) | 39(0.52%) | 0.937(0.672–1.307) | 0.920(0.659–1.285) |
| **Depressive disorder** | | | | p for interaction = 0.1597 |
| Mother | 1714(2.87%) | 213(2.82%) | 0.982(0.850–1.134) | 1.017(0.875–1.182) |

*(Continued)*

**Table 2.** (Continued)

| Factor | Control Group (N = 59809) | Kawasaki (N = 7566) | Crude OR (95% CI) | Adjusted OR (95% CI) |
|---|---|---|---|---|
| **Father** | 1534(2.56%) | 198(2.62%) | 1.021(0.879–1.186) | 1.058(0.905–1.237) |

Baseline period of parental disease is 2 years before the index date.

Fathers' data is not complete in the National Birth Reporting Database; therefore the number of fathers is 67 375.

## Discussion

KD is a rare systemic vasculitis condition and the leading cause of coronary artery abnormalities in children aged less than 5 years [1, 19, 20]. Our study investigated the association between parent characteristics and the incidence of KD in Taiwanese children. We revealed the link between parental (parental diseases during the previous 2 years) and perinatal factors and KD development. In Taiwan, the incidence of KD increased from 2005 to 2013; this may be attributed to the increase in urbanization and industrialization [21]. The finding of increasing KD cases with a declining growth rate is in accordance with that of a previous study [4]. A similar trend has been observed in North-East Asian countries, especially Japan and Korea [22, 23]. Nagao's study revealed that the total fertility rate is negatively associated with the incidence of KD [24].

Lower socioeconomic status, rural residence, and multiple-children families were associated with a lower risk of KD in our study. Hence, children with higher income, urban lifestyle, and belonging to single-child families are at higher risk of KD, leading to increasing KD cases in modern life [9–12, 24, 25]. The hygiene hypothesis first explained the etiology of atopic diseases [26]. According to the hypothesis, delayed exposure to infectious agents in childhood suppresses natural development and triggers the development of allergic diseases [27]. Moreover, it increases the risk of immune system diseases, including asthma, allergies, and KD. The incidence rates of such diseases have significantly increased since the 1960s [27]. The hygiene hypothesis also identified the environmental effect as a contributing factor to KD development through the dysregulation of early B cell development [25, 27]. This may explain our finding. The younger the parents are, the lower is the risk of having offspring with KD. This may be because the advanced age of parents leads to the accumulation of environmental factors, such as toxins and germline de novo variants [6, 28].

Older parents, especially in the age group of 35 to 45 years, had more offspring with KD. However, when parents were older than 45 years, OR no longer added value. This may be because of the small size of parents in this group. Huang et al. reported that maternal age less than 32 years may lead to fewer offspring with KD [6]. A higher maternal age is associated with a higher rate of KD with CAL and intravenous immunoglobulin resistance. Huang et al. suggested that this may be due to the maternal aging process, which may affect the development of offspring immune response [6]. No study has focused on the relationship between parental age and the risk of KD in offspring. Further investigation is required because of the trend of the increasing childbearing age. According to the theory of infectious triggers in genetically susceptible hosts of KD, siblings of children with KD are at a high risk of developing KD because of close contact [24, 29, 30]. In our study, the risk of KD was >4 times higher in these children than in children without siblings with KD.

Maternal autoimmune and allergic diseases may be associated with an increased risk of KD [5, 15, 31]. Several genes responsible for KD susceptibility have been identified, thereby proving the hereditary characteristics of KD [25, 32–35]. We evaluated the association

between maternal diseases (such as allergic and autoimmune diseases) and the risk of KD; however, no such association was noted. Therefore, more data are required to interpret these results. We discovered that paternal hypertension, hyperlipidemia, and chronic liver disease slightly increased the risk of KD in offspring; however, the increase was not statistically significant.

## Limitations

This study has a few limitations. First, the baseline period for assessing parental diseases was 2 years before the index patient was enrolled. The parental records were not comprehensive and detailed enough for evaluating the parental condition; therefore, some diseases may not have been tracked or recorded. For example, most symptoms of atopic dermatitis are in remission during adulthood [36], and rheumatic arthritis is always diagnosed after the age of 40 years [37]. Second, the siblings of children with KD had a >4 times higher risk of KD, but the chronology of and interval between events remains unknown. Third, the NHIRD has been updated until 2021. Medical technology has advanced considerably over the years in Taiwan, leading to more accurate diagnoses of Kawasaki disease. Additionally, the declining overall fertility rate in Taiwan has resulted in an increased proportion of families with only one child, and this shift in family structure may contribute to an increased prevalence of Kawasaki disease. The COVID-19 outbreak may have also affected the prevalence of Kawasaki disease. However, we could not evaluate the influence of these factors because our data only covered up to the year 2017. Accordingly, future research should consider using the most updated data to assess the effects of these factors on the risk of Kawasaki disease.

## Conclusions

We analyzed data from the NHIRD and NBRD to determine the incidence rate of KD and related risk factors to understand the etiology and pathogenesis of KD. Our findings, including the higher incidence rate of KD in children with an urban lifestyle, with siblings with KD, born to older mothers, and belonging to high-income and smaller families, are consistent with those of previous studies. However, no association between the risk of KD and maternal allergic and autoimmune diseases was observed.

## Supporting information

**S1 Checklist. Contains a comprehensive STROBE (Strengthening the Reporting of Observational Studies in Epidemiology) checklist tailored for the observational study presented in this research.** This checklist is intended to enhance transparency and completeness in the reporting of the study's design, methods, results, and discussion. The document details each criterion of the STROBE guidelines and elaborates on how our study meets these stipulations.
(DOCX)

## Acknowledgments

This study was partly based on data from the NHIRD provided by the NHI Administration, Ministry of Health and Welfare, and managed by the Health and Welfare Data Science Center (HWDC) in Taiwan. The interpretation and conclusions contained herein do not represent those of the NHI Administration, Ministry of Health and Welfare, or National Health Research Institutes.

## Author Contributions

**Conceptualization:** Chung-Fang Tseng, Chih-Jung Yeh, James Cheng-Chung Wei.

**Data curation:** Jing-Yang Huang, Chih-Jung Yeh, James Cheng-Chung Wei.

**Formal analysis:** Jing-Yang Huang, Chih-Jung Yeh, James Cheng-Chung Wei.

**Methodology:** Chung-Fang Tseng, Jing-Yang Huang, Chih-Jung Yeh, James Cheng-Chung Wei.

**Project administration:** Chih-Jung Yeh, James Cheng-Chung Wei.

**Supervision:** Chih-Jung Yeh, James Cheng-Chung Wei.

**Writing – review & editing:** Chung-Fang Tseng, Hsiao-Chen Lin, Chung-Yuh Tzeng.

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
