## [Decision Letter · Decision Letter 0]

26 Sep 2023

PONE-D-23-14951Association of Kawasaki disease with urbanization level and family characteristics in Taiwan: A nested case–control study using national-level dataPLOS ONE

Dear Dr. Yeh,

Thank you for submitting your manuscript to PLOS ONE. After careful consideration, we feel that it has merit but does not fully meet PLOS ONE’s publication criteria as it currently stands. Therefore, we invite you to submit a revised version of the manuscript that addresses the points raised during the review process.

We look forward to receiving your revised manuscript.

Kind regards,

Dong Keon Yon, MD, FACAAI, FAAAAI

Academic Editor

PLOS ONE

Journal Requirements:

"The Institutional Review Board of Chung Shan Medical University Hospital approved the study (CS19009)"

Additional Editor Comments:

Thank you for submitting your manuscript. The reviewers and I believe it is of potential value for our readers. However, the reviewers have raised a number of very important issues, and their excellent comments will need to be adequately addressed in a revision before the acceptability of your manuscript for publication in the Journal can be determined. We cannot guarantee that your revised paper will be chosen for publication; this would be solely based on how satisfactorily you have addressed the reviewer comments.

# All associations were considered statistically significant when P values were <.05. -> All associations were considered statistically significant when two-sided P values were <.05.

# We constructed the conditional logistic regression model to explore the association between baseline parental allergic disease and the risk of KD in children before the age of 5 years. -> Please cite the statistical paper (DOI: https://doi.org/10.54724/lc.2022.e3).

Reviewers' comments:

Reviewer's Responses to Questions

**Comments to the Author**

1. Is the manuscript technically sound, and do the data support the conclusions?

Reviewer #1: Yes

Reviewer #2: Yes

2. Has the statistical analysis been performed appropriately and rigorously? 

Reviewer #1: Yes

Reviewer #2: Yes

3. Have the authors made all data underlying the findings in their manuscript fully available?

Reviewer #1: Yes

Reviewer #2: Yes

4. Is the manuscript presented in an intelligible fashion and written in standard English?

Reviewer #1: Yes

Reviewer #2: Yes

5. Review Comments to the Author

Reviewer #1: This paper presents an investigation of the association of Kawasaki disease with urbanisation level and family characteristics in Taiwan using multi-dataset. The findings are interesting and may contribute to existing literature. The following issues need to be clarified before this reviewer has a favourable recommendation.

1. The latest NHIRD available would be for year 2021 or at least 2020. Please clarify why the latest data were not included, and how this is likely to impact the results drawn.

2. Confusion: in line 67, authors pointed out that the datasets used range from 2006 to 2015; however, in line 72, it was for 2003-2017.

3. It would be informative to provide readers who are not familiary the datasets (esp. NHIRD) with characteristics of the datasets. A vast majority of population is covered?

4. What is the reason(s) for cutpoints for some variables such as maternal age at delivery/father's age at delivery?

5. Was the multi-collinearity problem assessed? It seems to this reviewer that several comorbidities may exist at the same time and they may be stronly correlated.

Reviewer #2: This is a nation wide study of Kawasaki disease in Taiwan to investigate the cause of the disease. A family history of allergy, higher household income, smaller family size, and urbanization at birth were already reported in associated with incidence of KD. This paper confirmed the previous reports of Kawasaki disease. I agree the hygiene hypothesis to some extent, but your study did not mention about coronary involvement of the Kawasaki disease. I understand that this type of the study was very difficult to study the cause of coronary involvement, but please discuss the possible mechanism about higher maternal age and CAL.

6. PLOS authors have the option to publish the peer review history of their article (what does this mean?). If published, this will include your full peer review and any attached files.

Reviewer #1: **Yes: **Chih-Wei Pai

Reviewer #2: No

---

## [Author Response · Author response to Decision Letter 0]

7 Nov 2023

Dong Keon Yon, MD

Academic Editor

PLOS ONE

Oct. 23 2023

Dear Dr. Dong Keon Yon:

Re: Document reference No. PONE-D-23-14951

Please find attached a revised version of our document “Association of Kawasaki disease with urbanization level and family characteristics in Taiwan: A nested case–control study using national-level data”. We would like to resubmit for publication as a original article in PLOS ON

Your comments and those of the reviewers were highly insightful and enabled us to improve the quality of our document. In the following pages are our responses to each comment from the reviewer(s) as well as your own comments.

Revisions in the text are shown blue highlights. We hope that our revisions to the document combined with our accompanying responses will be sufficient to render our document suitable for publication in PLOS ONE.

Yours sincerely,

Chih-Jung Yeh

School of Public Health, Chung Shan Medical University

Tel.: 886 4 24730022 #12183

Fax: +886 4 23248179

E-Mail: alexyeh@csmu.edu.tw

Address: No. 110, Sec. 1, Jianguo N. Rd., South District, Taichung City 40201, Taiwan.

James Cheng-Chung Wei

Department of Allergy, Immunology & Rheumatology, Chung Shan Medical University Hospital

Tel.: +886 4 24739595 #34718

Fax: +886 4 24637389

E-Mail: jccwei@gmail.com

Address: No. 110, Sec. 1, Jianguo N. Rd., South District, Taichung City 40201, Taiwan.

Responses to the comments of Editor-in-Chief (If any)

# All associations were considered statistically significant when P values were <.05. -> All associations were considered statistically significant when two-sided P values were <.05.

Reply:

Thank you for your correction. We modified the original sentence of ‘All associations were considered statistically significant when P values were <.05.’ to 'All associations were considered statistically significant when two-sided P values were < .05.'

Modified sentence:

Line 142-144: “All statistical analyses were performed using SAS 9.4 (SAS Institute, Cary, NC, USA). A two-sided P value of <.05 indicated statistical significance.”

# We constructed the conditional logistic regression model to explore the association between baseline parental allergic disease and the risk of KD in children before the age of 5 years. -> Please cite the statistical paper (DOI: https://doi.org/10.54724/lc.2022.e3).

Reply:

Thank you very much for your suggestion. We added the reference (DOI: https://doi.org/10.54724/lc.2022.e3) to explain the regression analysis method.

Modified sentence:

Line 136-138: We constructed the conditional logistic regression model to explore the association between baseline parental allergic disease and the risk of KD in children before the age of 5 years [18].

Responses to the comments of Reviewer #1

Reviewer #1: This paper presents an investigation of the association of Kawasaki disease with urbanisation level and family characteristics in Taiwan using multi-dataset. The findings are interesting and may contribute to existing literature. The following issues need to be clarified before this reviewer has a favourable recommendation.

1. The latest NHIRD available would be for year 2021 or at least 2020. Please clarify why the latest data were not included, and how this is likely to impact the results drawn.

Reply: We initiated the application for NHIRD data authorization to the Health and Welfare Data Science Center in 2019, and the requested data covers the period from 2003 to 2017. NHIRD has been updated to the year 2021. However, based on our initial project proposal and the feasibility of data analysis, we are unable to utilize this latest data for our analysis. Future research can focus on analyzing the updated data to provide more meaningful scientific insights.

Modified sentence: 

Line 251-253: “Third, the NHIRD has been updated until 2021. However, based on our initial project proposal and the feasibility of data analysis, we are unable to utilize the latest data.”

2. Confusion: in line 67, authors pointed out that the datasets used range from 2006 to 2015; however, in line 72, it was for 2003-2017.

Reply: Sorry for the confusion, the wording was not clear and led to your misunderstanding. We selected the children born in Taiwan between January 1, 2006, and December 31, 2015, (n = 1 939 449) and observed their KD development before the age of 5 years. The current study utilized three nationwide databases in Taiwan, namely the NHIRD, the NBRD, and the National Death Index Database (NDID); Data from the NHIRD spanning from 2003 to 2017, the NBRD from 2006 to 2015, and the NDID from 2006 to 2017 were requested to identify baseline characteristics, including demographics, parental allergic diseases, and perinatal conditions, as well as the development of Kawasaki disease in offspring.

Modified sentence: 

Line 65-67: “For this retrospective nested case–control study, we selected children born in Taiwan between January 1, 2006, and December 31, 2015, (n = 1,939,449) and observed the development of KD before the age of 5 years in these children.”

and

Line 69-73: “We used three nationwide databases: the NHIRD (from 2003 to 2017), NBRD (from 2006 to 2015), and National Death Index Database (NDID, from 2006 to 2017). From the databases, baseline characteristics, namely demographic characteristics, parental allergic diseases, and perinatal conditions, and KD development in offspring, were obtained”

3. It would be informative to provide readers who are not familiary the datasets (esp. NHIRD) with characteristics of the datasets. A vast majority of population is covered?

Reply: Many thanks for your valuable suggestion, we added the information of datasets in line 73-78.

Modified sentence: 

Line 73-78: “The NHIRD is one of the largest and most comprehensive healthcare databases in the world [13] and includes medical claims data of >99% of Taiwan residents. Data on medical claims for inpatient and outpatient visits to clinics or hospitals (diagnosis codes and date of visit) were obtained from the NHIRD [13]. The incidence of maternal diseases (hypertension and diabetes) and childhood KD was obtained by reviewing medical claim records.”

4. What is the reason(s) for cutpoints for some variables such as maternal age at delivery/father's age at delivery?

Reply: Several studies have shown that advanced maternal age (35 years or older) is associated with higher risks of maternal morbidity, obstetric interventions, and adverse pregnancy outcomes (CMAJ. 2008 Jan 15; 178(2): 165–172). Previous research has also demonstrated that children born to mothers under the age of 25 or over the age of 35 exhibit poorer outcomes in terms of mortality, self-reported health, height, obesity, and the number of diagnosed conditions compared to those born to mothers aged 25–34 (Demography. 2012 Nov; 49(4): 10.1007/s13524-012-0132-x). Therefore, we categorized maternal/paternal age into four groups: <25, 25-34, 35-45, and ≥45 years old. The reasons for this categorization are provided in lines 81-87.

Modified sentence: 

Line 81-87: “Advanced maternal age (≥35 years) has been associated with higher risks of maternal morbidity, obstetric interventions, and adverse pregnancy outcomes[14]. A study demonstrated that children born to mothers aged <25 or >35 years had poorer outcomes in terms of mortality, self-reported health, height, obesity, and the number of diagnosed conditions than those born to mothers aged 25–34 years [15]. Therefore, we categorized maternal/paternal age into four groups: <25, 25–34, 35–45, and ≥45 years old.”

5. Was the multi-collinearity problem assessed? It seems to this reviewer that several comorbidities may exist at the same time and they may be stronly correlated.

Reply: We have evaluated the multicollinearity problem in the logistic regression analysis. The regression model does not exhibit severe multicollinearity. In Table 2, we included the crude odds ratios (OR) for offspring's Kawasaki disease associated with various parental comorbidities, as well as the adjusted odds ratios (aOR) after controlling for baseline demographics and perinatal conditions. Although the crude OR results indicate significant associations between paternal allergic rhinitis, hypertension, hyperlipidemia, chronic liver disease, and offspring's Kawasaki disease, all the aOR values are similar to the crude OR. This also suggests there is a very mild multicollinearity problem when estimating the association between offspring's Kawasaki disease and paternal comorbidities. 

Modified sentence: 

1.We added the crude ORs in Table 2.

Table 2. Results of Conditional Multivariable Logistic Regression Analysis for Risk Factors Among Children with Kawasaki Disease

Factor Control Group (N= 59 809) Kawasaki (N= 7566) Crude OR

(95% CI) Adjusted OR

(95% CI)

Allergy disease 

Atopic dermatitis p for interaction=0.2668

Mother 1119(1.87%) 141(1.86%) 0.996(0.835-1.189) 0.991(0.828-1.186)

Father 303(0.51%) 37(0.49%) 0.965(0.686-1.359) 0.905(0.630-1.300)

Allergic rhinitis p for interaction=0.7031

Mother 5044(8.43%) 647(8.55%) 1.015(0.932-1.106) 1.003(0.913-1.102)

Father 4536(7.58%) 622(8.22%) 1.092(1.000-1.191) 1.072(0.973-1.181)

Asthma p for interaction=0.3905

Mother 871(1.46%) 123(1.63%) 1.118(0.924-1.353) 1.089(0.893-1.329)

Father 834(1.39%) 100(1.32%) 0.947(0.769-1.167) 0.896(0.719-1.117)

Hypertension p for interaction=0.2078

Mother 606(1.01%) 83(1.10%) 1.048(0.891-1.234) 1.127(0.882-1.441)

Father 1642(2.75%) 238(3.15%) 1.151(1.002-1.321) 1.137(0.987-1.309)

Hyperlipidemia p for interaction=0.5155

Mother 385(0.64%) 42(0.56%) 0.862(0.626-1.186) 0.873(0.620-1.229)

Father 2025(3.39%) 294(3.89%) 1.154(1.018-1.307) 1.125(0.991-1.277)

Chronic liver disease p for interaction=0.4180

Mother 1268(2.12%) 168(2.22%) 1.048(0.891-1.234) 1.044(0.872-1.251)

Father 4146(6.93%) 575(7.60%) 1.104(1.009-1.209) 1.085(0.988-1.192)

Diabetes p for interaction=0.6786

Mother 690(1.15%) 76(1.00%) 0.869(0.685-1.103) 0.850(0.666-1.085)

 Father 693(1.16%) 90(1.19%) 1.027(0.823-1.281) 1.028(0.821-1.288)

COPD p for interaction=0.3199

Mother 241(0.40%) 35(0.46%) 1.149(0.805-1.639) 1.065(0.730-1.554)

Father 387(0.65%) 52(0.69%) 1.063(0.795-1.421) 1.001(0.740-1.355)

Autoimmune disease 

Diffuse diseases of connective tissue p for interaction=0.9299

Mother 341(0.57%) 43(0.57%) 0.997(0.725-1.370) 0.971(0.705-1.336)

Father 86(0.14%) 10(0.13%) 0.919(0.477-1.770) 0.957(0.495-1.853)

Rheumatoid arthritis and other inflammatory polyarthropathies p for interaction=0.9403

Mother 70(0.12%) 11(0.15%) 1.243(0.658-2.347) 1.230(0.649-2.331)

Father 67(0.11%) 13(0.17%) 1.535(0.847-2.781) 1.530(0.842-2.782)

Ankylosing spondylitis and other inflammatory spondylopathies p for interaction=0.9192

Mother 101(0.17%) 14(0.19%) 1.096(0.626-1.918) 1.105(0.630-1.938)

Father 329(0.55%) 39(0.52%) 0.937(0.672-1.307) 0.920(0.659-1.285)

Depressive disorder p for interaction=0.1597

Mother 1714(2.87%) 213(2.82%) 0.982(0.850-1.134) 1.017(0.875-1.182)

Father 1534(2.56%) 198(2.62%) 1.021(0.879-1.186) 1.058(0.905-1.237)

2. Line 185-187: “The crude ORs indicated significant associations of paternal allergic rhinitis, hypertension, hyperlipidemia, and chronic liver disease with offspring KD, and the values of the adjusted ORs were similar to the crude ORs”.

Responses to the comments of Reviewer #2

Reviewer #2: This is a nation wide study of Kawasaki disease in Taiwan to investigate the cause of the disease. A family history of allergy, higher household income, smaller family size, and urbanization at birth were already reported in associated with incidence of KD. This paper confirmed the previous reports of Kawasaki disease. I agree the hygiene hypothesis to some extent, but your study did not mention about coronary involvement of the Kawasaki disease. I understand that this type of the study was very difficult to study the cause of coronary involvement, but please discuss the possible mechanism about higher maternal age and CAL.

Replay: Thank you to reviewers for affirming the conclusions of our study. We did not discuss about coronary artery invasion in Kawasaki disease in our study. This was not included in this designed experiment. In line 223-225 of our article: Huang et al. reported that maternal age less than 32 years may lead to fewer offspring with KD (Pediatr Rheumatol Online J. 2019;17(1):46. doi: 10.1186/s12969-019-0348-z). A higher maternal age is associated with a higher rate of KD with CAL and intravenous immunoglobulin resistance. 

 In previous paper (Pediatr Rheumatol Online J. 2019;17(1):46. doi: 10.1186/s12969-019-0348-z) described that “KD is reported to be a disease with autoimmune-like or autoinflammatory response that is triggered in genetically predisposed subjects by certain environmental factors. Growing evidence has shown a link between KD and immune-mediated diseases, especially allergic diseases or autoimmune diseases. Children with allergic diseases like urticaria, allergic rhinitis, and atopic dermatitis are subsequently at an increased risk of KD. In contrast, a subsequent risk for allergic diseases including asthma, allergic rhinitis, and atopic dermatitis has been found to be increased in patients with a history of KD. As for immune-mediated diseases, previous studies have indicated associations with increasing maternal age at delivery with an increased risk for type 1 diabetes and food allergies in the offspring. A fine-tuned balance between genetic, immunological, metabolic, and hormonal factors is necessary for reproduction, and all of these factors are likely involved in the aging process. Therefore, maternal aging may affect health outcomes in offspring, including with regard to the development of their immune response. “

I agree the above possible mechanism, of course, I also spent a lot of time trying used “Pubmed of national library medicine” search for the possible mechanism about high maternal age and KD with CAL. But I did not figure out other possible mechanism.

Modified sentence: 

Line:228-230: “Huang et al. suggested that this may be due to the maternal aging process, which may affect the development of offspring immune response [6]”

---

## [Decision Letter · Decision Letter 1]

21 Nov 2023

PONE-D-23-14951R1Association of Kawasaki disease with urbanization level and family characteristics in Taiwan: A nested case–control study using national-level dataPLOS ONE

Dear Dr. Yeh,

Thank you for submitting your manuscript to PLOS ONE. After careful consideration, we feel that it has merit but does not fully meet PLOS ONE’s publication criteria as it currently stands. Therefore, we invite you to submit a revised version of the manuscript that addresses the points raised during the review process. Please submit your revised manuscript by Jan 05 2024 11:59PM. If you will need more time than this to complete your revisions, please reply to this message or contact the journal office at plosone@plos.org. Please include the following items when submitting your revised manuscript:A rebuttal letter that responds to each point raised by the academic editor and reviewer(s). You should upload this letter as a separate file labeled 'Response to Reviewers'.A marked-up copy of your manuscript that highlights changes made to the original version. You should upload this as a separate file labeled 'Revised Manuscript with Track Changes'.An unmarked version of your revised paper without tracked changes. You should upload this as a separate file labeled 'Manuscript'.

We look forward to receiving your revised manuscript.

Kind regards,

Dong Keon Yon, MD, FACAAI, FAAAAI

Academic Editor

PLOS ONE

Journal Requirements:

Additional Editor Comments:

Please address minor comments from the reviewers.

Reviewers' comments:

Reviewer's Responses to Questions

**Comments to the Author**

1. If the authors have adequately addressed your comments raised in a previous round of review and you feel that this manuscript is now acceptable for publication, you may indicate that here to bypass the “Comments to the Author” section, enter your conflict of interest statement in the “Confidential to Editor” section, and submit your "Accept" recommendation.

Reviewer #1: (No Response)

Reviewer #2: (No Response)

2. Is the manuscript technically sound, and do the data support the conclusions?

Reviewer #1: Yes

Reviewer #2: Yes

3. Has the statistical analysis been performed appropriately and rigorously? 

Reviewer #1: Yes

Reviewer #2: Yes

4. Have the authors made all data underlying the findings in their manuscript fully available?

Reviewer #1: Yes

Reviewer #2: Yes

5. Is the manuscript presented in an intelligible fashion and written in standard English?

Reviewer #1: Yes

Reviewer #2: Yes

6. Review Comments to the Author

Reviewer #1: The authors did a good job by responding to this reviewer's previous comments and questions. My final question/comment is that the latest NHIRD was not analysed. Please clarify, in the respect of Kawasaki disease with urbanization level and family characteristics in Taiwan, how this is likely to impact the results drawn.

Reviewer #2: This is a nationwide study of Kawasaki disease. This manuscript is well revised. Please change Yoshio (Line 209) to Nagao in the text.

7. PLOS authors have the option to publish the peer review history of their article (what does this mean?). If published, this will include your full peer review and any attached files.

Reviewer #1: **Yes: **Chih-Wei Pai

Reviewer #2: No

---

## [Author Response · Author response to Decision Letter 1]

13 Dec 2023

Dong Keon Yon, MD

Academic Editor

PLOS ONE

Oct. 23 2023

Dear Dr. Dong Keon Yon:

Re: Document reference No. PONE-D-23-14951

Please find attached a revised version of our document “Association of Kawasaki disease with urbanization level and family characteristics in Taiwan: A nested case–control study using national-level data”. We would like to resubmit for publication as a original article in PLOS ON

Your comments and those of the reviewers were highly insightful and enabled us to improve the quality of our document. In the following pages are our responses to each comment from the reviewer(s) as well as your own comments.

Revisions in the text are shown blue highlights. We hope that our revisions to the document combined with our accompanying responses will be sufficient to render our document suitable for publication in PLOS ONE.

Yours sincerely,

Chih-Jung Yeh

School of Public Health, Chung Shan Medical University

Tel.: 886 4 24730022 #12183

Fax: +886 4 23248179

E-Mail: alexyeh@csmu.edu.tw

Address: No. 110, Sec. 1, Jianguo N. Rd., South District, Taichung City 40201, Taiwan.

James Cheng-Chung Wei

Department of Allergy, Immunology & Rheumatology, Chung Shan Medical University Hospital

Tel.: +886 4 24739595 #34718

Fax: +886 4 24637389

E-Mail: jccwei@gmail.com

Address: No. 110, Sec. 1, Jianguo N. Rd., South District, Taichung City 40201, Taiwan.

Responses to the comments of Reviewer

6. Review Comments to the Author

Please use the space provided to explain your answers to the questions above. You may also include additional comments for the author, including concerns about

dual publication, research ethics, or publication ethics. (Please upload your review as an attachment if it exceeds 20,000 characters)

Reviewer #1: The authors did a good job by responding to this reviewer's previous comments and questions. My final question/comment is that the latest NHIRD was not analysed. Please clarify, in the respect of Kawasaki disease with urbanization level and family characteristics in Taiwan, how this is likely to impact the results drawn.

Reply: One limitation of our study is that we utilized NHIRD data spanning from 2003 to 2017, while NHIRD has been updated to include data up to 2021. Over the years, medical technology has made significant strides in Taiwan, leading to more accurate diagnoses of Kawasaki disease. Additionally, the declining total fertility rate has resulted in a higher proportion of single-child families, which could potentially contribute to an increased prevalence of Kawasaki disease. Furthermore, the outbreak of COVID-19 may also impact the prevalence of Kawasaki disease. However, due to the scope of our data ending in 2017, we were unable to evaluate the influence of these factors. Future research should consider utilizing the most recent data to assess the effects of these factors on the risk of Kawasaki disease.

Modified sentence:

Line 252-259 “Medical technology has advanced considerably over the years in Taiwan, leading to more accurate diagnoses of Kawasaki disease. Additionally, the declining overall fertility rate in Taiwan has resulted in an increased proportion of families with only one child, and this shift in family structure may contribute to an increased prevalence of Kawasaki disease. The COVID-19 outbreak may have also affected the prevalence of Kawasaki disease. However, we could not evaluate the influence of these factors because our data only covered up to the year 2017. Accordingly, future research should consider using the most updated data to assess the effects of these factors on the risk of Kawasaki disease..”

Reviewer #2: This is a nationwide study of Kawasaki disease. This manuscript is well revised. Please change Yoshio (Line 209) to Nagao in the text.

Reply: We greatly appreciate your valuable suggestion. We have made the necessary change from "Yoshio" to "Nagao" in line 209 to ensure the accuracy of the information presented. Once again, we sincerely thank you for your valuable input.

Modified sentence: 

Line 209:” Yoshiro’s Nagao’s study revealed that the total fertility rate is negatively associated with the incidence of KD”

---

## [Decision Letter · Decision Letter 2]

15 Dec 2023

Association of Kawasaki disease with urbanization level and family characteristics in Taiwan: A nested case–control study using national-level data

PONE-D-23-14951R2

Dear Dr. Yeh,

We’re pleased to inform you that your manuscript has been judged scientifically suitable for publication and will be formally accepted for publication once it meets all outstanding technical requirements.

Kind regards,

Dong Keon Yon, MD, FACAAI, FAAAAI

Academic Editor

PLOS ONE

Additional Editor Comments (optional):

This is an excellent paper.

Reviewers' comments:

Reviewer's Responses to Questions

**Comments to the Author**

1. If the authors have adequately addressed your comments raised in a previous round of review and you feel that this manuscript is now acceptable for publication, you may indicate that here to bypass the “Comments to the Author” section, enter your conflict of interest statement in the “Confidential to Editor” section, and submit your "Accept" recommendation.

Reviewer #1: All comments have been addressed

2. Is the manuscript technically sound, and do the data support the conclusions?

Reviewer #1: Yes

3. Has the statistical analysis been performed appropriately and rigorously? 

Reviewer #1: Yes

4. Have the authors made all data underlying the findings in their manuscript fully available?

Reviewer #1: Yes

5. Is the manuscript presented in an intelligible fashion and written in standard English?

Reviewer #1: Yes

6. Review Comments to the Author

Reviewer #1: All my previous comments have been addressed properly. I have no further comment. Thank you. Well done.

7. PLOS authors have the option to publish the peer review history of their article (what does this mean?). If published, this will include your full peer review and any attached files.

Reviewer #1: **Yes: **Chih-Wei Pai

---

## [Editor Report · Acceptance letter]

25 Dec 2023

PONE-D-23-14951R2 

PLOS ONE

Dear Dr. Yeh, 

I'm pleased to inform you that your manuscript has been deemed suitable for publication in PLOS ONE. Congratulations! Your manuscript is now being handed over to our production team.

Kind regards, 

on behalf of

Dr. Dong Keon Yon 

Academic Editor

PLOS ONE